# Monitoring the Quality of Life and the Relationship between Quality of Life, Dietary Intervention, and Dietary Adherence in Patients with Coeliac Disease

**DOI:** 10.3390/nu16172964

**Published:** 2024-09-03

**Authors:** Eszter Dakó, Sarolta Dakó, Veronika Papp, Márk Juhász, Johanna Takács, Éva Csajbókné Csobod, Erzsébet Pálfi

**Affiliations:** 1Health Sciences Division, Doctoral College, Semmelweis University, Üllői Str. 26, 1085 Budapest, Hungary; dako.eszter@phd.semmelweis.hu; 2Department of Surgery, Transplantation and Gastroenterology, Semmelweis University, Üllői Str. 26, 1085 Budapest, Hungary; dako.sarolta@semmelweis.hu (S.D.); papp.veronika@semmelweis.hu (V.P.); 3Gastro Clinic, Bokor Str. 17-21, 1037 Budapest, Hungary; juhaszmarkdr@gmail.com; 4Department of Social Sciences, Faculty of Health Sciences, Semmelweis University, Üllői Str. 26, 1085 Budapest, Hungary; takacs.johanna@semmelweis.hu; 5Department of Dietetics and Nutritional Sciences, Faculty of Health Sciences, Semmelweis University, Üllői Str. 26, 1085 Budapest, Hungary; csajbokne.csobod.eva@semmelweis.hu

**Keywords:** coeliac, gluten-free diet, quality of life, treatment adherence and compliance, dietary behaviors, dietary patterns

## Abstract

Inadequate adherence to a gluten-free diet in coeliac disease triggers autoimmune reactions and can reduce the quality of life. The strict diet requires constant vigilance, which can cause psychological distress. Our research aimed to assess the quality of life in adult patients with coeliac disease and to find a correlation between quality of life, dietary intervention, and adherence. The study included 51 adult patients with coeliac disease who completed a quality-of-life questionnaire. Adherence was assessed using serological tests and a dietary adherence test. The patients were divided into two groups: those on a gluten-free diet for at least three months (Group I) and newly diagnosed patients (Group II). Group I showed a significant decrease in the dysphoria subscale of the quality-of-life test between the first and last surveys. Poor quality of life was associated with worse adherence in Group II. A higher “Health concerns” quality of life subscale score was also associated with worse adherence in Group II. Our results suggest that dietetic care may be beneficial for patients with coeliac disease by reducing dysphoria. We recommend regular and long-term dietary monitoring from diagnosis to ensure adherence to a gluten-free diet and to maintain quality of life.

## 1. Introduction

Coeliac disease (CD) is a systemic autoimmune disease caused by gluten intake, based on genetic predisposition, for which the only treatment is a lifelong gluten-free diet (GFD) [1]. Even small amounts of gluten can initiate the autoimmune process, inhibiting the regeneration of the mucous membrane. A precise quantification of this amount is challenging due to the heterogeneity in available research, which encompasses studies with varying objectives and designs. Key variables contributing to this difficulty include differences in the duration of interventions and the lack of standardization in gluten dosing protocols. Gluten at a dose of 50 mg/day is already considered harmful for patients with coeliac disease based on the relevant literature [2,3]. A study from 2023 found that a daily intake of 6 mg of gluten was associated with a 0.2% risk of relapse and 1.5 g with a 100% risk of relapse [4]. Coeliac disease, being a chronic disease, places a heavy burden on the patient as well as on the healthcare system. Care of patients with coeliac disease should be provided by a multidisciplinary healthcare team involving physicians, nurses, dietitians, and psychologists [5]. The task of the healthcare team is to assess adherence and the quality of life of patients and, if necessary, increase it.

Identifying the factors affecting the quality of life and understanding the relationship between the factors can greatly contribute to successful treatment. Many factors, such as age, clinical manifestations, comorbidities, sexual dysfunction, availability of gluten-free products, knowledge about the disease, and dietary adherence, can influence the quality of life of patients with coeliac disease [6,7,8]. Some studies report significant associations between physical activity and quality of life, as well as adherence to a gluten-free diet [9]. In addition to the general quality of life tests, special methods for assessing the quality of life related to coeliac disease have also emerged. However, only a few prospective studies have been conducted using these methods. Currently, the relationship between quality of life and diet adherence by patients with coeliac disease is not yet fully understood [7,8,10]. A strict gluten-free diet usually results in clinical and microscopic remission and improvement in quality of life and nutritional status [11]. However, as a result of inadequate adherence, autoimmune reactions can cause further damage and persisting symptoms. Thus, the quality of life does not improve. Additionally, the quality of life does not improve with a GFD in all cases [12,13]. Many patients with coeliac disease report that the lack of alternative treatments to replace diet has a negative impact on their quality of life, and/or they continue to experience negative psychological symptoms such as anxiety [12,13,14]. Furthermore, like other chronic diseases, coeliac disease also presents challenges that also affect the quality of life. Following a strict diet can be difficult for patients [12]. Gluten-free products differ from traditional ones in consistency, color, and taste. These differences may reduce enjoyment of the meal [8]. Dietary restrictions require constant attention, which can cause psychological stress and sometimes lead to failure or temporary suspension of the diet. Even though a wide range of gluten-free products are available nowadays, purchasing diet food can still be a problem due to the high prices. This is especially true for products with a more optimal composition, such as products with a higher fiber and lower added sugar content [11]. Therefore, a strict GFD is a social and financial burden for patients and their families [15].

According to international professional guidelines, regular medical and dietary monitoring is recommended for patient care. Adherence to a gluten-free diet requires a high level of knowledge and motivation facilitated by dietary education. During the first year of following a gluten-free diet, close monitoring contributes to developing and maintaining good dietary adherence, acquiring balanced gluten-free eating habits, and providing psychological support when necessary [1]. The healthcare team responsible for patient care is tasked with preventing or managing complications, assessing and improving adherence, and enhancing the quality of life in patients.

Our research aimed to assess and monitor the quality of life and diet adherence of adult patients with coeliac disease cared for in outpatient care and to discover the correlations between the examined parameters and dietary intervention to improve patient care.

## 2. Materials and Methods

### 2.1. Patient Population

Implementing quarantine and other restrictions associated with the ongoing pandemic significantly impeded the progress of our follow-up clinical study. As a result, we were forced to close patient recruitment earlier than expected. The study included 51 adult patients (aged 18 years and over) with coeliac disease (gold standard certified coeliac disease)—13 males (25.5%) and 38 females (74.5%). The mean age of participants was 35.92 years (SD = 11.74) with a minimum age of 20 years and a maximum age of 70 years.

Pregnancy, cancer, chronic alcohol or drug consumption, pacemaker implantation, and the presence of a prosthesis were exclusion criteria. Prostheses were an exclusion criterion mainly due to the body composition analysis criteria. Body composition analysis criteria are not part of this publication but are an important pillar of our research.

During data processing, the patients were divided into two groups according to whether they had been following the gluten-free diet for at least three months (Group I) or were newly diagnosed (Group II). In the sample of 51 people, 19 patients (37.3%) were newly diagnosed, including 5 men and 14 women. The previously diagnosed group included 32 people (62.7%), 8 men and 24 women.

### 2.2. Study Design

Patient enrolment was continuous between 2018 and 2020. In addition to the cross-sectional study, we planned a one-year follow-up. Participation in the research was voluntary, preceded by oral and written information, and informed written consent was obtained from all the patients before participation in the study.

The study protocol included four meetings for newly diagnosed patients and three for previously diagnosed patients already following a gluten-free diet, a summary of which can be seen in Table 1. We planned to examine dietary adherence and quality of life for each patient three times, but this was not possible for everyone due to the disease prevention and control measures implemented during the COVID-19 pandemic.

### 2.3. Quality of Life Assessment

The Coeliac Disease Quality of Life (CD-QoL) is a self-administered questionnaire, validated in several languages, designed to assess coeliac disease-specific quality of life. The questionnaire consists of 20 items, each of which can be assessed using a 5-point Likert scale response. Respondents choose the option that best corresponds with their feelings about the statement or question. Response options are listed in ascending order: Not at all is 1 point and A great deal is 5 points. It contains four clinically relevant subscales: ‘constraints/limitations’ (9 items), ‘dysphoria’ (4 items), ‘health concerns’ (5 items), and ‘inappropriate treatment’ (2 items). The ‘limitations’ scale assesses limitations due to coeliac disease, such as difficulties in social interactions, social stigma, and travel difficulties. The ‘dysphoria’ subscale reflects the overload, fear, and depressed state caused by the disease, while the ‘health concerns’ subscale evaluates concerns about health damage and complications caused by coeliac disease. The subscale for inadequate treatment measures how satisfied the patient is with the treatment options. The total score can be expressed on a scale of 20–100. A higher score indicates worse coeliac disease-specific quality of life [16]. The first survey was carried out during the first consultation for patients already on a diet for a longer time (V1). For the newly diagnosed, it was carried out on the second dietary consultation when they were already on a gluten-free diet (V1). The data were collected on paper in person or online during the consultations

### 2.4. Assessment of Dietary Adherence

Adherence is a term proposed by the World Health Organization (WHO) to describe patient cooperation. Dietary adherence is the concept of following a prescribed medical diet. Contrary to the previously used term of compliance, the concept of adherence presupposes the patient as an active participant in the treatment/recovery [17].

Serological tests, including IgA-tissue transglutaminase (tTG-IgA), endomysial antibody (EMA), and deamidated gliadin peptide antibodies (DGP), have poor sensitivity for detecting ongoing but low-level gluten exposure and persistent intestinal damage [18,19,20]. Several studies of gluten exposure did not show a significant change in tTG-IgA levels over time compared to placebo. In studies of newly diagnosed patients, both EMA and tTG usually normalize despite the presence of histological damage and/or gluten exposure [19].

Adherence was assessed using a 7-question test (Coeliac Dietary Adherence Test, CDAT) developed by Leffler et al. (2009), which is a simple and clinically sensitive tool for monitoring dietary adherence. This questionnaire assesses four aspects of adherence to a GFD—coeliac symptoms, self-efficacy, reasons to follow a GFD, and perceived adherence to a GFD [12,19]. The questions can be answered using a scale from 1 to 5 (1 = never/completely agree, 5 = always/completely disagree). The total score is obtained by summing up the answers to the items. The minimum possible score is 7 and the maximum is 35. The lower the score, the stricter the adherence to the diet, and the higher the score, the worse the adherence [12,21].

An important pillar of the care of patients with coeliac disease and another way to assess dietary adherence, but at the same time not sensitive enough in itself, is the serological examination of patients (tTG-IgA monitoring), which, according to our research protocol, was recorded at every patient consultation [19].

### 2.5. Statistical Analysis

During descriptive statistical analyses, relative frequency (%), mean (M), and standard deviation (SD) were determined. Differences between the two examined groups (Group I and Group II) based on age were tested using the independent samples *t*-test. Fisher’s exact test was used to test the association between gender and the groups.

The quality of life (CD-QoL) scores were assessed at three time points during the research (V1, V2, V3). ANOVA was used to examine the differences in the measured CD-QoL scores across the three time points considering the group’s factorial repeated measures. In the model, CD-QoL scores measured at V1, V2, and V3 were used as within-subjects variables (time main effect with Bonferroni confidence interval adjustment), and the groups (Group I vs. Group II) were used as between-subject factors for Time × Group interaction effect analysis. In the case of significant interaction and because of the small sample size, simple main effects analysis was also used to compare Time main effects for each level of the between-subjects factor. Partial eta-squared (η^2^_p_) was reported as an effect size measurement.

The association between the CD-QoL results obtained during the last (third) visit of the study and dietary adherence (CDAT results) was examined using Spearman’s rank-order correlation.

Statistical analyses were performed using IBM SPSS Statistics for Windows, Version 25.0 program (IBM Corp. Released 2017, Armonk, NY, USA). The level of significance was set at 0.05.

### 2.6. Ethics

The research was carried out with the approval of the Semmelweis University Regional and Institutional Committee of Science and Research Ethics (RKEB). The ethics clearance number is SE RKEB-7/2018.

## 3. Results

The majority of patients enrolled in the study were female (74.5%), with a mean age of 34.5 (SD = 12.0) and 38.4 (SD = 11.2) years in the two groups, respectively. Age did not show a statistically significant difference between the groups (t(49) = 1.150, *p* = 0.256), and the female/male proportions were not different between the two groups (χ^2^(1,N = 51) = 0.011, *p* = 0.917). The main parameters of the two groups of patients are shown in Table 2.

### 3.1. CD-QoL Scores

The participants, i.e., 51 people, completed a specific test on the quality of life of people living with coeliac disease. A total of 43 people answered the test at least two times and 31 at least three times, according to the following distribution: QoL 1: n = 51 (Group I: 32 people, Group II: 19 people), QoL 2: n = 43 (Group I: 27 people, Group II: 16 people), QoL 3: n = 31 (Group I: 21 people, Group II: 10 people). A summary of the results of the QoL tests can be found in Table 3.

The 31 patients with three CD-QoL results were included in the factorial repeated measures analysis of variance. The results show that neither the total QoL score nor the QoL subscales were significantly different between the results obtained at the time of the V1, V2, and V3 visits, hence the change was not significant (Time main effect). The groups (Group I vs. Group II) did not significantly affect the results either, thus the interaction effect was not significant. As shown in Table 4, the simple main effects analysis shows that for the dysphoria subscale, there is a significant overall difference between the results obtained at the three time points in Group I (F (2, 40) = 3.759, *p* = 0.032, η^2^_p_ = 0.16).

During the multiple comparisons (post hoc) in Group I, a statistically significant difference was observed between the results of the first and third assessments, while the second analysis showed an intermediate value (Figure 1). In Group II, there were no significant differences between the three assessments (Table 4).

### 3.2. CDAT Results

The results of the adherence assessment (CDAT3) conducted during the V3 visit are shown in Table 5. The CDAT score can range from 7 to 35, with a lower score indicating better adherence. For Group I, the mean time since diagnosis at the time of the CDAT3 test was 4.5 years (54 months), with a minimum of 1.25 years (15 months) and a maximum of 10.6 years (128 months). Subjects in Group II were on a gluten-free diet for 12 months at the time of the CDAT3 test.

At the last measurement point (V3) of the follow-up research, our patients completed the CD-QoL test for the third time, and the relationship between the results of the CD-QoL test and dietary adherence assessment, taken at the same time, was examined in both groups.

In the group of newly diagnosed patients included in the research (Group II), a statistically significant, positive moderate correlation was revealed between the QoL total score and CDAT tests completed during the third visit (rho(8) = 0.669, *p* = 0.034). No significant association was found (rho(11) = 0.262, *p* = 0.387) in Group I (Figure 2).

The “Health Concern” subscale and the CDAT test at visit three showed a strong statistically significant positive correlation (rho(8) = 0.826, *p* = 0.003) in Group II, while no significant correlation was found in Group I (rho(11) = −0.063, *p* = 0.839) (Figure 3).

## 4. Discussion

### 4.1. Quality of Life Assessment

The results in Figure 1 showing a statistically significant difference between the dysphoria subscale V1 and V3 scores for patients with a previous diagnosis (Group I) can be interpreted as meaning that, since the score for assessment 3 is lower, the third consultation showed a significant improvement in the condition of these patients compared to the first consultation in terms of depression and overload due to coeliac disease.

It should be noted that these results could be seen in patients who already knew their condition and were following a gluten-free diet at the time of enrollment. In their case, thorough nutrition and dietary care may have had an impact on coping with the disease burden, even though they were not new to coeliac disease. In addition to teaching the basic principles of the diet, the dietician can help patients customize their diet and manage different life situations appropriately, so they can be considered competent professionals in the development of self-management of the patients. A systematic problem-solving method ensuring a high standard of professional work in the field of nutrition is the Nutrition Care Process and Model (NCPM), of which nutrition monitoring and evaluation is an essential element in addition to nutrition assessment, nutrition diagnosis, and nutrition intervention [22,23]. Dietetic practices can also be more effective in the care of patients with coeliac disease if there is an opportunity for multiple consultations, i.e., follow-up of the patients.

Our research did not include measuring physical activity (PA) levels using standardized instruments. However, it is important to note that physical activity is a significant factor in the quality of life [9,24]. A survey indicated that adherence to a gluten-free diet may influence physical activity levels in patients with coeliac disease positively [9]. Given that physical activity benefits the quality of life, it seems reasonable to posit that dietary intervention may also influence physical activity and quality of life by enhancing adherence. Studies indicate that regular physical activity improves mood, reduces anxiety, and enhances mental health [24,25]. Group sports also provide social interactions, which can reduce feelings of isolation among patients with inflammatory bowel disease (IBD) [25]. This may also prove beneficial for patients with coeliac disease who frequently report feelings of isolation due to the limitations imposed by their condition [9].

### 4.2. Dietary Adherence

Table 5 shows that there is no significant difference between the CDAT test results measured at the last 12-month consultation of the study (V3) in the two groups. The results for both groups can be considered good, as the total score for the CDAT questionnaire can range from 7 to 35—the lower the value, the better the assumed adherence. A review of the literature reveals that COVID-19 can also have a detrimental impact on therapeutic adherence by patients with chronic illnesses such as inflammatory bowel disease [26]. Considering these findings, it seems reasonable to hypothesize that the pandemic, which emerged during our study, may have had some impact on dietary adherence by patients with coeliac disease. However, a systematic review reported that in the case of coeliac disease, diet adherence and health-related quality of life remained relatively stable before and during the pandemic [27]. In developing a strategy to improve diet adherence by patients with coeliac disease, assessing the potential risk of dietary non-adherence during the pandemic on a case-by-case basis may be beneficial.

Among the newly diagnosed patients (Group II) who entered the research and were monitored for 12 months (n = 10), four patients still had positive serological (tTG-IgA) results after one year of being on the diet. Of the 13 patients in Group I who were successfully followed throughout, 5 had positive serological results based on the laboratory test performed on the third visit. According to the group assignment criteria, patients who had not started dieting at the time of enrollment but had been eating gluten-free foods for at least 3 months were considered ‘old patients’, thus everyone in this group had been following the gluten-free diet for at least 15 months at the time of the third visit. However, even with a proper diet, serology values do not always stabilize over short periods; instead, it usually takes one year, although in some cases it may take up to two years [20]. In the case of those patients who still had positive serology at the 1-year limit as defined in the European Society for the Study of Coeliac Disease (ESsCD) 2019 guidelines, it is important that, even if there is no deliberate dietary error and there is adequate motivation to follow the diet, a dietitian be consulted to detect possible accidental gluten intake [1]. In this research, 4 out of 11 patients who had been on a diet for at least 1.5 years and one out of eight patients who had been on a diet for at least two years still had tTG-IgA levels in the positive range.

### 4.3. Association between CDAT and QoL

Based on the literature, the relationship between adherence and quality of life after starting a gluten-free diet cannot be predicted in advance due to a lack of evidence [10,12]. Finding and maintaining a balance between them would be particularly important to increase the quality of life of patients with coeliac disease.

Based on the results of our study (Figure 2), a lower QoL score (i.e., better quality of life) is associated with a lower CDAT score (i.e., better adherence) in the group of newly diagnosed patients who had been on a gluten-free diet for 12 months (Group II). Studies also indicate that following a gluten-free diet can be associated with an improvement in the quality of life, which is consistent with our observation in the newly diagnosed group [12].

Based on our results, the higher ‘Health Concerns’ QoL subscale score, which indicates stronger concern, is associated with a higher CDAT score (i.e., worse adherence) in the group of patients who were newly diagnosed and had been on a gluten-free diet for 12 months (Figure 3); thus, we can see that fear of disease complications and health impairments, which is are important factors affecting the quality of life of people living with coeliac disease, is not associated with more disciplined dietary adherence, but with worse adherence in Group II. An important factor when interpreting the results is that these patients are in their first gluten-free year of life, which is a particularly difficult period, as in this year, they have to face the diagnosis of a chronic illness. Acceptance and processing can be a long process, so initially, denial and grief can have a bad effect on dietary adherence. At the same time, they also have to learn the basics of a lifelong strict diet, which can be challenging in itself [10]. In addition, besides facing everyday difficulties in the first year, they also usually encounter several special life situations, such as holiday meals and trips. Additionally, possible symptoms do not disappear overnight, serology may improve at varying rates, body composition may not necessarily change in the desired direction, and it may take longer for relatives or acquaintances to accept the new life situation. In a worse case, patients may also face stigma when, for example, eating outside the home; thus, the diet is a huge psychological, social, and financial burden to the patient that does not necessarily bring a quick and spectacular improvement in health status. The importance of dietetic care also lies in increasing and maintaining adherence despite the above factors, and for the specialist to support the patient as a kind of coach, in addition to reducing their concerns, in maintaining dietary adherence in the long term [22,23].

### 4.4. Strengths and Limitations

The main strength of this clinical research is the monitoring of the quality of life among histologically confirmed coeliac cases and the examination of its relationship with dietary intervention and adherence. However, our research also has limitations. Our sample has a relatively small number of cases; therefore, the results are not representative of the Hungarian coeliac population. Generalizing the results at an international level is also difficult because of the different situations faced by patient populations in different countries. Even within an individual country, there may be variations in patients’ education levels, the possibility of purchasing specialty foods, and the state of the healthcare system.

Further research with a large number of cases is recommended to obtain a more detailed, reliable picture of the factors that determine the quality of life of patients with coeliac disease and the relationships between them in different age groups and cultural backgrounds, thereby enabling targeted, professional, and complex care for those affected.

## 5. Conclusions

Medical and dietary control play an important role in the complex care of patients with coeliac disease. Our results confirm the phenomenon experienced in practice, that even patients who have been on a diet for a long time may need dietary intervention to achieve a balanced diet and reduce the various challenges and dysphoria associated with the disease.

The results obtained during our research and the data from specialized literature sources allow us to conclude that proper adherence to diet is associated with specific quality of life related to coeliac disease. In our sample, better quality of life was associated with better dietary adherence in patients who had been on a diet for 1 year. According to our results, health-related concerns may be associated with reduced dietary adherence in adults who have been on a diet for a year. It is of particular importance that patients should not be excluded from the health system after dietary education, and instead, dietary adherence and quality of life should be monitored over the long term (years).

## Figures and Tables

**Figure 1 nutrients-16-02964-f001:**
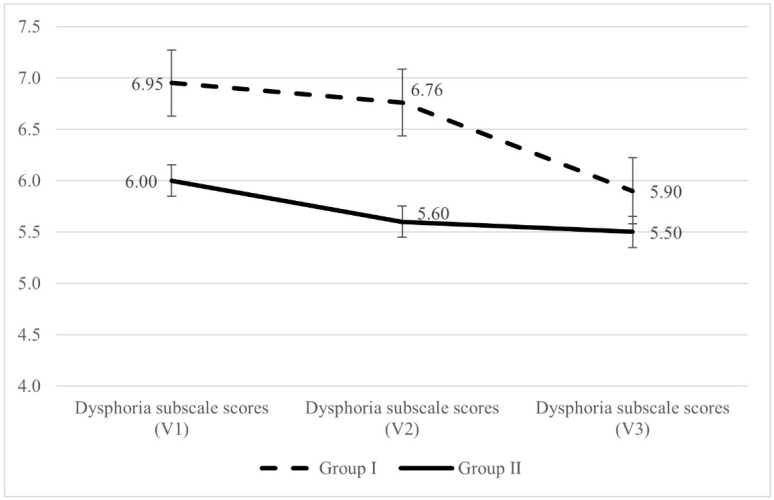
Results of the dysphoria subscale at the three measurement times based on group (error bar: standard error), n (Group I) = 21, n (Group II) = 10. The dysphoria subscale score can range from 4 to 20.

**Figure 2 nutrients-16-02964-f002:**
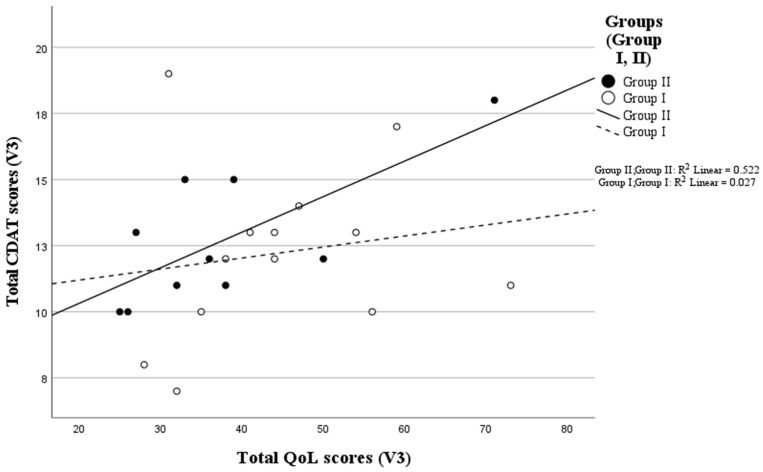
Relationship between total CD-QoL and total CDAT scores at the V3 visit in the study groups. n (Group I) = 13, n (Group II) = 10.

**Figure 3 nutrients-16-02964-f003:**
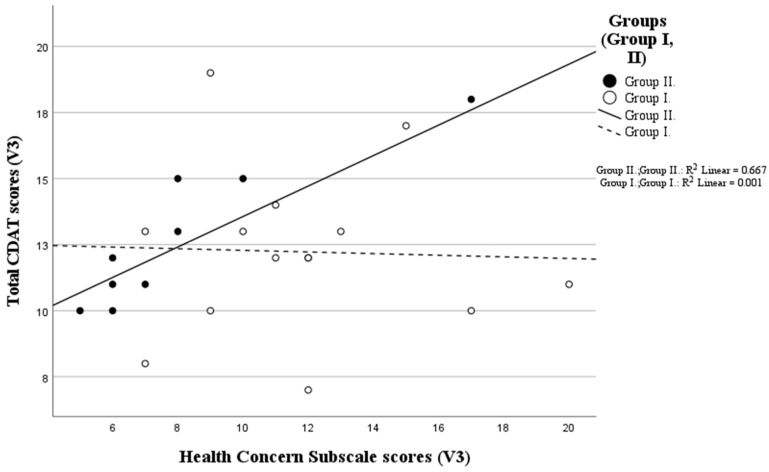
Relationship between total CD-QoL health concern subscale and total CDAT scores at the V3 visit in the study groups. n (Group I) = 13, n (Group II) = 10.

**Table 1 nutrients-16-02964-t001:** Research protocol (CDAT—Coeliac Dietary Adherence Test, CD-QoL—Coeliac Disease Quality of Life, tTG—tissue transglutaminase, EMA/DGP—endomysium antibody/deamidated gliadin peptide antibody, x/-—intervention planned/not planned for current visit).

Group I	Inclusion (n = 32) (Informed Consent/Screening)		Visit 1 V1	Visit 2 V2	Visit 3 V3
Dietetic intervention			x	x	x
CDAT			x	x	x
CD-QoL			x	x	x
Serology (tTG)			x	x	x
**Group II**	**Inclusion (n = 19)** **(Informed Consent/Screening)**	**Visit 0** **(GFD for** **0 month)** **V0**	**Visit 1** **(GFD for** **3 month)** **V1**	**Visit 2** **(GFD for** **6 month)** **V2**	**Visit 3** **(GFD for** **12 month)** **V3**
Dietetic intervention		x	x	x	x
CDAT		-	x	x	x
CD-QoL		-	x	x	x
Serology (tTG)		x	x	x	x
Serology (EMA/DGP)		x	-	-	-

**Table 2 nutrients-16-02964-t002:** The main characteristics of the studied groups.

	Group I	Group II
n (%)	32 (62.7)	19 (37.3)
Age (years), mean ± SD	34.5 ± 12.0	38.4 ± 11.2
Female, n (%)	24 (75.0)	14 (73.7)
Years since CD diagnosis, mean ± SD (min–max)	5.2 ± 6.3 (0.25–23)	-

**Table 3 nutrients-16-02964-t003:** CD-QoL test scores for the studied groups.

Groups		n	Minimum	Maximum	M	SD
Group I	QoL 1 total score	32	28	71	43.28	11.86
	QoL 2 total score	27	25	79	41.7	12.74
	QoL 3 total score	21	28	73	41.14	11.29
Group II	QoL 1 total score	19	26	78	43.63	13.08
	QoL 2 total score	16	24	69	40.56	11.83
	QoL 3 total score	10	25	71	37.7	13.87

**Table 4 nutrients-16-02964-t004:** Results of the simple main effects analysis of QoL.

Group I (n = 21)										
	V1		V2		V3		F	*p*	η^2^_p_ *	Post Hoc
	M	SD	M	SD	M	SD				
QoL (Total)	44.24	11.69	41.90	13.13	41.14	11.29	2.017	0.146	0.09	
Limitations	22.71	5.77	21.33	6.42	20.52	6.26	3.236	0.050	0.14	
Dysphoria	6.95	3.07	6.76	3.55	5.90	2.36	3.759	0.032	0.16	V1 = V2, V1 > V3, V2 = V3
Health Concerns	10.24	4.07	10.05	3.58	10.57	3.71	0.541	0.586	0.03	
Inadequate Treatment	6.10	2.12	6.00	1.61	5.38	1.20	1.943	0.157	0.09	
**Group II (n = 10)**										
	**V1**		**V2**		**V3**		**F**	** *p* **	**η^2^_p_ ***	**Post hoc**
	**M**	**SD**	**M**	**SD**	**M**	**SD**				
QoL (Total)	40.90	10.83	40.80	11.42	37.70	13.87	1.395	0.273	0.13	
Limitations	20.70	6.65	21.50	6.21	20.50	7.61	0.262	0.772	0.03	
Dysphoria	6.00	2.06	5.60	2.22	5.50	2.46	0.654	0.532	0.07	
Health Concerns	10.10	4.70	9.50	3.69	8.50	3.66	0.827	0.453	0.08	
Inadequate Treatment	7.00	2.67	6.80	1.75	6.10	1.66	0.639	0.539	0.07	

* η^2^_p_ = effect size (partial eta-squared).

**Table 5 nutrients-16-02964-t005:** Results of adherence assessment and serological tests in the study groups conducted at the V3 visit.

CDAT3—Total Score						Serology (tTG-IgA)	
	n	Minimum	Maximum	M	SD	Negative	Positive
Group I	13	7	19	12.23	3.30	8	5
Group II	10	10	18	12.7	2.58	6	4

## Data Availability

Records and data pertaining to this study are stored electronically at the Semmelweis University, Faculty of Health Sciences, Department of Dietetics and Nutritional Sciences, Hungary, and can be provided by the corresponding author on a reasonable request.

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
