# Peer review of "Monitoring the Quality of Life and the Relationship between Quality of Life, Dietary Intervention, and Dietary Adherence in Patients with Coeliac Disease"

_nutrients, 2024, doi:10.3390/nu16172964_

Round 1

Reviewer 1 Report

Comments and Suggestions for Authors

Coeliac disease is a chronic condition that inevitably impacts the quality of life (QoL) of affected patients, and, consequently, examining its determinants that may lead to variations in QoL is scientifically appropriate and relevant.

I read this manuscript with interest, but I believe that there are some aspects that need further expansion. The QoL in chronic digestive conditions is a very broad topic with many determinants that, in my opinion, should be discussed. Recent cross-sectional studies conducted in the context of long-term gluten-free diets have evaluated that, despite this, erectile and sexual dysfunction in patients with coeliac disease is a relevant issue. This should be mentioned in the introduction.

In addition, beyond proper nutrition, another determinant of QoL that is currently gaining significant attention in the scientific literature is regular physical activity (PA). There are studies that have examined PA in patients with coeliac disease, which should be discussed (https://pubmed.ncbi.nlm.nih.gov/35603827/). It would also be useful to compare these findings with studies in similar chronic gastrointestinal conditions, such as IBD. In this regard, a brief comparison with recent BE-FIT-IBD studies would be beneficial.

There is no mention of the impact of COVID-19. Your study discusses therapeutic adherence (which, in the case of coeliac disease, certainly refers to the diet). In chronic gastrointestinal conditions, a previous SARS-CoV-2 infection has drastically impacted therapeutic adherence in patients with chronic digestive diseases, such as IBD, where it has emerged as a predictor (https://pubmed.ncbi.nlm.nih.gov/35973931/). It would be useful to describe this aspect and compare it with data from coeliac settings.

I would recommend rewriting the statistical analysis paragraph in a more organized and technical manner.

In Table 2, a P-value for comparison between the two groups is missing to assess sample heterogeneity.

Author Response

Thank you very much for taking the time to review this manuscript. Please find the detailed responses below and the corresponding revisions/corrections highlighted changes in the re-submitted files.

Comment 1: I read this manuscript with interest, but I believe that there are some aspects that need further expansion. The QoL in chronic digestive conditions is a very broad topic with many determinants that, in my opinion, should be discussed. Recent cross-sectional studies conducted in the context of long-term gluten-free diets have evaluated that, despite this, erectile and sexual dysfunction in patients with coeliac disease is a relevant issue. This should be mentioned in the introduction.

Response 1: Thank you for pointing this out. We have, accordingly, supplemented the introduction.

“Many factors, such as age, clinical manifestations, comorbidities, sexual dysfunction, availability of gluten-free products, knowledge about the disease, and dietary adherence, can influence the quality of life of coeliac patients [6,7,8].” (Page 2, Line 52-55)

6. Romano, L.; Pellegrino, R.; Sciorio, C.; Barone, B.; Gravina, A.G.; Santonastaso, A.; Mucherino, C.; Astretto, S.; Napolitano, L.; Aveta, A., et al. Erectile and sexual dysfunction in male and female patients with celiac disease: A cross-sectional observational study. Andrology 2022, 10, 910-918, doi:10.1111/andr.13186.

Comment 2: In addition, beyond proper nutrition, another determinant of QoL that is currently gaining significant attention in the scientific literature is regular physical activity (PA). There are studies that have examined PA in patients with coeliac disease, which should be discussed (https://pubmed.ncbi.nlm.nih.gov/35603827/).

Response 2: Thank you for bringing this to our attention. We agree with the comment and have made the following changes in the manuscript:

“Some studies report significant associations between physical activity and quality of life, as well as adherence to a gluten-free diet [9].” (Page 2, Line 55-56)

“Our research did not include measuring physical activity (PA) levels using standardized instruments. However, it is important to note that physical activity is a significant factor in the quality of life [9,24]. A survey has indicated that adherence to a gluten-free diet may positively influence physical activity levels in patients with coeliac disease [9]. (Page 8, Line 265-268)

9. Bouery, P.; Attieh, R.; Sacca, L.; Sacre, Y. Assessment of the social quality of life and the physical activity of adult celiac disease patients following a gluten-free diet in Lebanon. Nutr Health 2024, 30, 103-113, doi:10.1177/02601060221095685.

Comment 3: It would also be useful to compare these findings with studies in similar chronic gastrointestinal conditions, such as IBD. In this regard, a brief comparison with recent BE-FIT-IBD studies would be beneficial.

Response 3: Thank you for this comment. We have, accordingly, supplemented the discussion.

“Our research did not include measuring physical activity (PA) levels using standardized instruments. However, it is important to note that physical activity is a significant factor in the quality of life [9,24]. A survey has indicated that adherence to a gluten-free diet may positively influence physical activity levels in patients with coeliac disease [9]. Given that physical activity benefits the quality of life, it seems reasonable to posit that dietary intervention may also influence physical activity and quality of life by enhancing adherence. Studies indicate that regular physical activity improves mood, reduces anxiety, and enhances mental health [24,25]. Group sports activities also provide social interaction, which can reduce feelings of isolation among patients with inflammatory bowel disease (IBD) [25]. This may also prove beneficial for patients with coeliac disease, who frequently report feelings of isolation due to the limitations imposed by their condition [9].”  (Page 8, Line 265-276)

24. Gravina, A.G.; Pellegrino, R.; Durante, T.; Palladino, G.; D'Onofrio, R.; Mammone, S.; Arboretto, G.; Auletta, S.; Imperio, G.; Ventura, A., et al. Inflammatory bowel diseases patients suffer from significant low levels and barriers to physical activity: The "BE-FIT-IBD" study. World J Gastroenterol 2023, 29, 5668-5682, doi:10.3748/wjg.v29.i41.5668.

25. Stafie, R.; Singeap, A.M.; Rotaru, A.; Stanciu, C.; Trifan, A. Bridging the gap: Unveiling the crisis of physical inactivity in inflammatory bowel diseases. World J Gastroenterol 2024, 30, 1261-1265, doi:10.3748/wjg.v30.i10.1261.

Comment 4: There is no mention of the impact of COVID-19. Your study discusses therapeutic adherence (which, in the case of coeliac disease, certainly refers to the diet). In chronic gastrointestinal conditions, a previous SARS-CoV-2 infection has drastically impacted therapeutic adherence in patients with chronic digestive diseases, such as IBD, where it has emerged as a predictor (https://pubmed.ncbi.nlm.nih.gov/35973931/). It would be useful to describe this aspect and compare it with data from coeliac settings.

Response 4:  Thank you for this comment. As the research was designed before the emergence of the SARS-CoV-2 virus, it was impossible to plan to examine the pandemic's effects. Nevertheless, the pandemic was indeed an essential factor. We have, accordingly, supplemented the discussion.

“A review of the literature reveals that the disease COVID-19 can also have a detrimental impact on the therapeutic adherence of patients with chronic illnesses, such as inflammatory bowel disease [26]. Considering these findings, it seems reasonable to hypothesise that the pandemic, which emerged during our study, may have had some impact on dietary adherence among patients with coeliac disease. However, a systematic review reported that in the case of coeliac disease, diet adherence and health-related quality of life remained relatively stable before and during the pandemic [27]. In developing a strategy to improve the diet adherence of patients with coeliac disease, assessing the potential risk of dietary non-adherence during the pandemic period on a case-by-case basis may be beneficial.” (Page 9, Line 281-291)

Comment 5: I would recommend rewriting the statistical analysis paragraph in a more organized and technical manner.

Response 5: Thank you for this comment. We organized a statistical analysis paragraph based on the structure of the results section and provided more details following your comment. Based on the further comments, we also added further statistical analysis.

“During descriptive statistical analyses, the relative frequency (%), mean (M), and standard deviation (SD) were shown. The differences between the two examined groups (Group I and Group II) on age were tested using the independent samples t-test. Fisher’s exact test was used to test the association between gender and the groups.

The quality of life (CD-QoL) scores were assessed three times during the research (V1, V2, V3). ANOVA was used to examine the differences between the three times measured CD-QoL considering the group's factorial repeated measures. In the model, CD-QoL measured at V1, V2, and V3 was used as a within-subjects variable (Time main effect with Bonferroni confidence interval adjustment), and the groups (Group I vs Group II) were used as a between-subjects factor for Time x Group interaction effect. In the case of significant interaction and because of the small sample size, simple main effects analysis was also used to compare Time main effects for each level of the between-subjects factor. Partial eta-squared (η2p) was reported as an effect size measurement.” (Page 4, Line 164-176)

Comment 6: In Table 2, a P-value for comparison between the two groups is missing to assess sample heterogeneity.

Response 6: Thank you, we added this information in-text.

“Age did not show a statistically significant difference between the groups (t(49) = 1.150, p = 0.256), and female:male proportions were not different between the two groups (χ2(1,N=51) = 0.011, p = 0.917).”
(Page 5, Line 189-192)

Additional clarifications

In addition, we have provided the necessary clarifications in the revised manuscript:

1. In the abstract, we have replaced the word 'celiac' with 'coeliac' to maintain consistent spelling throughout the manuscript.

2. We have corrected the following section in Table 2.:

Female, n (%)   Group I: 24 (75,0)          Group II: 14 (73,7)

3. We have replaced the phrase ‘with’ with ‘used’ in line 180

Reviewer 2 Report

Comments and Suggestions for Authors

The article addresses an interesting topic for Nutrients readers, namely the relationship between quality of life and dietary adherence in patients with celiac disease.

The research is well written and follows all the steps of an original article.

The introduction, despite being well constructed, well-founded (12 bibliographic references) and updated (the obsolescence index of the bibliographic citations, or in other words, the median age of the citations is 5 years), is excessively short, which does not allow us to focus on the current state of the subject.

The methodology is correct, but due to the small sample size, both the results and the conclusions may be compromised (it would be interesting to indicate whether a prior estimation of the sample size was made). We believe it would be interesting to increase the sample.  In formal aspects, the methodology used allows us to achieve the proposed objectives.

The results are well described and are supported by enlightening tables and figures that facilitate reading and understanding. The results are in agreement with the proposed objectives.

The discussion, in spite of its length, is supported by very few articles (only 5) and also with a very high obsolescence rate of 15 years, so we consider that it should be expanded with new studies that are also more current. It is not normal for the discussion to be based on fewer studies than the introduction.

The conclusions are concrete, accurate and reflect very well the most outstanding results.

The bibliography is very scarce and not very up to date in general, since the overall obsolescence rate is 8 years, especially due to the discussion. No self-citations are observed

Author Response

Thank you very much for taking the time to review this manuscript. Please find the detailed responses below and the corresponding revisions/corrections highlighted changes in the re-submitted files.

Comment 1: The introduction, despite being well constructed, well-founded (12 bibliographic references) and updated (the obsolescence index of the bibliographic citations, or in other words, the median age of the citations is 5 years), is excessively short, which does not allow us to focus on the current state of the subject.

Response 1:  Thank you for your feedback. We have expanded the introduction accordingly, hoping that this will help readers to engage more deeply and focus more effectively on the subject. (Page 1, Line 39-43; Page 2, Line 46-47, 49-52; 78-85)

Comment 2: The methodology is correct, but due to the small sample size, both the results and the conclusions may be compromised (it would be interesting to indicate whether a prior estimation of the sample size was made). We believe it would be interesting to increase the sample. The results are well described and are supported by enlightening tables and figures that facilitate reading and understanding. The results are in agreement with the proposed objectives.

Response 2:  Thank you for this comment. With this study design (number of groups: 2, number of measurements: 3), based on a sample size calculation with 0.25 effect size f, alpha=0.05, power=0.95, the required total sample size would be 44, which was respected in the present study. At the same time, there are further confounding factors, further essential characteristics of the participants (not only age and gender, which did not show differences between the groups), etc., which can influence the results. We included these in the limitation section.

We have included additional information in the Material and Methods section. 

“Implementing quarantine and other restrictions associated with the ongoing pandemic has significantly impeded the progress of our follow-up clinical study. As a result, we were forced to close patient recruitment earlier than expected.” (Page 2, Line 91-93)

Comment 3: The discussion, in spite of its length, is supported by very few articles (only 5) and also with a very high obsolescence rate of 15 years, so we consider that it should be expanded with new studies that are also more current. It is not normal for the discussion to be based on fewer studies than the introduction.

Response 3: Thank you for your observation. We have expanded the list of references that support the discussion with additional, more recent articles, complementing the previously relevant sources:

  • 9. Bouery, P.; Attieh, R.; Sacca, L.; Sacre, Y. Assessment of the social quality of life and the physical activity of adult celiac disease patients following a gluten-free diet in Lebanon. Nutr Health 2024, 30, 103-113, doi:10.1177/02601060221095685.
  • 10. McDermid, J.M.; Almond, M.A.; Roberts, K.M.; Germer, E.M.; Geller, M.G.; Taylor, T.A.; Sinley, R.C.; Handu, D. Celiac Disease: An Academy of Nutrition and Dietetics Evidence-Based Nutrition Practice Guideline. J Acad Nutr Diet 2023, 123, 1793-1807 e1794, doi:10.1016/j.jand.2023.07.018.
  • 24. Gravina, A.G.; Pellegrino, R.; Durante, T.; Palladino, G.; D'Onofrio, R.; Mammone, S.; Arboretto, G.; Auletta, S.; Imperio, G.; Ventura, A., et al. Inflammatory bowel diseases patients suffer from significant low levels and barriers to physical activity: The "BE-FIT-IBD" study. World J Gastroenterol 2023, 29, 5668-5682, doi:10.3748/wjg.v29.i41.5668.
  • 25. Stafie, R.; Singeap, A.M.; Rotaru, A.; Stanciu, C.; Trifan, A. Bridging the gap: Unveiling the crisis of physical inactivity in inflammatory bowel diseases. World J Gastroenterol 2024, 30, 1261-1265, doi:10.3748/wjg.v30.i10.1261.
  • 26. Pellegrino, R.; Pellino, G.; Selvaggi, F.; Federico, A.; Romano, M.; Gravina, A.G. Therapeutic adherence recorded in the out-patient follow-up of inflammatory bowel diseases in a referral center: Damages of COVID-19. Dig Liver Dis 2022, 54, 1449-1451, doi:10.1016/j.dld.2022.07.016.
  • 27. Amirian, P.; Zarpoosh, M.; Moradi, S.; Jalili, C. Celiac disease and COVID-19 in adults: A systematic review. PLoS One 2023, 18, e0285880, doi:10.1371/journal.pone.0285880.

Comment 4: The bibliography is very scarce and not very up to date in general, since the overall obsolescence rate is 8 years, especially due to the discussion.

Response 4:  Thank you for pointing this out. We have expanded the bibliography with seven additional recent references to provide a more up-to-date foundation, particularly for the discussion section. In addition to the new literature included in the discussion section, we have supplemented the manuscript with the following reference:

6.           Romano, L.; Pellegrino, R.; Sciorio, C.; Barone, B.; Gravina, A.G.; Santonastaso, A.; Mucherino, C.; Astretto, S.; Napolitano, L.; Aveta, A., et al. Erectile and sexual dysfunction in male and female patients with celiac disease: A cross-sectional observational study. Andrology 2022, 10, 910-918, doi:10.1111/andr.13186.

Additional clarifications

In addition, we have provided the necessary clarifications in the revised manuscript:

  1. In the abstract, we have replaced the word 'celiac' with 'coeliac' to maintain consistent spelling throughout the manuscript.
  2. We have corrected the following section in Table 2.:

Female, n (%)   Group I: 24 (75,0)          Group II: 14 (73,7)

  1. We have replaced the phrase ‘with’ with ‘used’ in line 180

Round 2

Reviewer 1 Report

Comments and Suggestions for Authors

The authors augmented the discussion of the topic in the manuscript and provided a timely point-by-point response by providing appropriate revisions to the manuscript. I believe that, in this form, the manuscript is worthy of publication.

Reviewer 2 Report

Comments and Suggestions for Authors

Is ok